# Squeezed from the top: "Social Outburst" (2019) and elite overproduction. A study of the dynamics of Chilean political instability from the approach of Structural Demographic Theory

**Manuel Muñoz-Rodríguez**[1,2]*, **Rosana Ferrero**[2], **Juan Pablo Luna**[2,3,4,5], **Mauricio Lima**[1,2]

**1** Departamento de Ecología, Pontificia Universidad Católica de Chile, Santiago, Chile, **2** Center of Applied Ecology and Sustainability (CAPES), Pontificia Universidad Católica de Chile, Santiago, Chile, **3** Escuela de Gobierno e Instituto de Ciencia Política, Pontificia Universidad Católica de Chile, Santiago, Chile, **4** Instituto Milenio Fundamentos de los Datos, Santiago, Chile, **5** Instituto Milenio VIODEMOS, Santiago, Chile

\* munozrodriguez.manuel@gmail.com

**Data Availability Statement:** All relevant data are within the manuscript and its Supporting Information files.

## Abstract

On October 18, 2019, Chile experienced the most important social upheaval since the country regained democracy in the late 1980s. The "Social Outbreak" surprised economic and political elites and seemed paradoxical to the international community who had often praised Chile as a model of successful development. In this paper, we used structural-demographic theory to analyze the interaction between the overproduction of elites and the stagnation in the relative income of the population as the underlying structural cause of Chilean political instability. This theory was able to predict the three most significant instances of political tension in the recent history of Chile: the crisis of the late 1960s that culminated in the coup d'état of 1973, popular mobilizations during the 1980s, and the recent student mobilizations and social upheaval. Our results suggest that, at least during the period 1938–2019, Chilean sociopolitical dynamics is determined by the same structural drivers.

## Introduction

Understanding the factors that underlie the dynamics of political instability in modern societies is essential to identifying the structural underpinnings (and thus, long-term drivers) of contemporary political events associated with the crisis of contemporary liberal democracies [1–3]. In this sense, the structural demographic theory (SDT) represents an interpretive framework that contributes to understanding the relationship between demographic and economic processes and their potential causal link with general well-being, intra-elite conflicts, and internal peace in a social system [3, 4]. The basic notion of SDT is that sociopolitical instability stems from tensions resulting from imbalances between population, social structures, and resources [1]. In this context, political pressure is exerted when the requirements or expectations of social groups exceed the structural limits of a specific socioeconomic environment [5],

**Funding:** Center for Applied Ecology and Sustainability (CAPES), ANID PIA/BASAL FB0002 MM received a grant: ANID Advanced Human Capital Program for funding this research (https://www.conicyt.cl/becasconicyt/2014/08/28/beca-doctorado-nacional-2015/) The funders had no role in study design, data collection and analysis, decision to publish, or preparation of the manuscript.

**Competing interests:** The authors have declared that no competing interests exist.

increasing the intensity of intra- and intergroup competition, and rising the probability that specific factors will lead to that the latent conflict escalates rapidly, threatening the stability of the political system [1–3]. Therefore, SDT does not seek to encompass all the complexity associated with the phenomenon of political conflict in social systems, nor does it ignore the importance of the agency of political actors. It only allows us to predict, through the interaction of structural variables, periods when the collective mood is more likely to trigger major scenarios of institutional instability.

From the SDT perspective, the probability of political instability increases when a certain society experiences the following conditions: 1) *State Fiscal Distress* (SFD), as a consequence of an imbalance in the income/expense budget and/or loss of institutional legitimacy (increased mistrust), which affects the capacity of the State to fulfill its functions and obligations; 2) increase in the *Elite Mobilization Potential* (EMP), derived from a greater tension on the resources captured by this sector, which generates fragmentation within this social group and alienation from the State among the dominant sectors of society (intra-elite competition); and 3) an increase in the *Mass Mobilization Potential* (MMP), caused by a relative drop in the living standards of the population and an increase in the social patterns that predispose a population to mobilization, such as an increase in youth or the urbanization of the population [1–3]. Through the estimation of the *Political Stress Indicator* (PSI) *(PSI = SFD x MMP x EMP)*, different empirical studies have used the SDT framework to analyze crises and state collapses in agrarian, industrial and post-industrial societies, finding an important relationship between PSI dynamics and the pattern of emergence of political instability events [1–4, 6].

According to the theoretical framework of SDT, one of the main drivers of sociopolitical instability is the anti-phase dynamic between the growth in the number of people who aspire to occupy elite positions and the proxies for the well-being of the population. Different empirical studies have found that there is a greater probability of the emergence of events of political violence when the well-being of the population is low and there is an overproduction of elites (a number greater than what a given society can sustain) [2–4]. In this sense, exploring the relationship between these two variables (in a long wave) can contribute to better understanding processes involved in increasing social tension.

On October 18, 2019, Chile experienced its most significant social upheaval since the country regained democracy in the late 1980s. Protests staged by high school students in Santiago abruptly unleashed a disruptive force that spread throughout the national territory, generating multiple spontaneous mobilizations and riots that escalated the level of the conflict and jeopardized institutional stability. In only three months (October to December 2019), 3,300 protest actions were registered, according to data from the Conflict Observatory (COES), an amount nine times greater than the total number of actions reported in the entire previous decade [7]. These mobilizations, which were called "Social Outbreak", surprised Chile's political and social elites and seemed paradoxical to the international community, who had often praised Chile as a model of successful development. Indeed, during the last 25 years, Chile has experienced a high rate of economic growth, a doubling of real per capita income, and a reduction in poverty levels from 38.6% of the population to approximately 8% [8]. Similarly, Chileans' life expectancy at birth has experienced a marked improvement due to a notable reduction of infant mortality [9].

In the last 80 years, Chile has experienced other events of political instability of great relevance. The most important of them, the crisis of the late 60s and early 70s, which culminated with a Coup d'état (1973) that represented a profound break with the political and economic model that prevailed until then. This event meant the installation of a dictatorial regime (1973–1989) that was responsible for the death and disappearance of more than 3,000 victims [10]. On the other hand, during the Latin American debt crisis (beginning of the 80s), an

important cycle of civil mobilizations erupted in Chile, which constituted the main challenge faced by the Pinochet dictatorship [10, 11].

This study uses the framework of Structural Demographic Theory (SDT) to identify possible structural pressures involved with the "Chilean Social Outburst", carrying out an empirical reconstruction of the tensions accumulated in the different social groups (elites and population) for the period 1938–2019, evaluating the potential interaction between them. Based on this and considering the trajectories of the structural variables (SFD, MMP and EMP), the dynamics of the Political Stress Indicator (PSI) was estimated and its predictive capacity on the trajectory of political instability observed in Chile in the last 80 years was evaluated.

## Materials and methods

### Political instability index observed in Chile (1865–2019)

To reconstruct the dynamics of political instability in Chile (1938–2019) (S1 Fig), we searched for social conflict events in more than 30 secondary sources, including digital files, press articles, official documents and books (all collected events, their description and source information are shown in detail in the database available in the supplementary materials). Based on this collection of events, an indicator was estimated, following the categorization of Banks et al., (2021) [12, see http://www.databanksinternational.com]. The calculated indicator (the index of observed political instability) represents a weighted measure of the annual frequency of the following conflicts: anti-government demonstrations, riots, political assassinations, general strikes, terrorist attacks/clashes between armed political factions, governance crises, purges and revolutions/coups [12]. To standardize the specific weight of each event within its historical context, the relative frequency was estimated for certain periods. For this, the historical macro division was based on Vial (2006) and Rodriguez (2018) [13, 14] weighting the events by category and standardizing them by the frequency of conflicts registered in three periods: Mesocratic Republic (1937–1973), Military Dictatorship (1973–1989) and Return to Democracy (1990–2019). In this sense, the index of political instability observed in a particular year is given by the formula:

$$index_t = \sum_{i=1}^{8} \frac{frequency_{(t,i)}}{total\ ocurrence_{(p,i)}} \tag{1}$$

Where "t" corresponds to time in years (1865–2019), "p" is the historical period (Mesocratic Republic 1938–1973, Military Dictatorship 1973–1989 and Democracy 1990–2019) and "i" refers to each of the 8 categories of political conflict in which the collected events are classified (anti-government demonstrations, riots, political assassinations, general strikes, terrorist attacks/clashes between armed political factions, governance crises, purges and revolutions/coups).

### Structural demographic theory (SDT) (1938–2019)

To estimate the dynamics of the political stress index (PSI), the series of the three structural variables were reconstructed: fiscal stress (SFD), population mobilization potential (MMP) and intra-elite tension (EMP), following the model proposed by Turchin (2013, 2016) [2, 3].

**SFD.** According to Turchin (2013, 2016) [2, 3] the fiscal stress indicator (SFD) consists of two components, one economic and another related to the level of institutional distrust. Regarding the first component, two indicators were used. One of them based on the public expenditure/public income ratio, which represents a measure of fiscal balance at the central government level (*B.fis*), and a second indicator based on the path of relative public debt (stock) relative to GDP (*Y/G*). To estimate *B.fis*, data on public expenditure and income (period 1938–

2010) from Diaz et al., (2016) [15] were used. For the period 2010–2019, the time series (period 1990–2019) of the Dipres (National Budget Office of the Government of Chile) was used [16]. Both series were spliced by linear interpolation, using as coefficient the average rate of change between series for the overlapping period (1990–2010). A similar methodology was followed for the Y/G component, where data from Diaz et al 2016 [15] was used for the period 1938–2010 and spliced with the Ministry of Finance of Chile series (1991–2019) [17].

As for the institutional confidence component, the "Contract Intensive Money indicator" (CIM) was used as a proxy, which represents an indirect estimator of institutional legitimacy, based on a measure of the degree of confidence in compliance with contracts and in the right to property [18].The CIM indicator represents a measure of the amount of savings in the economy in relative terms, being estimated from the ratio between the difference between the total money supply (M2) and circulating money (coins and bills) and M2 (*M2-circulating/M2*) [18]. The data on M2 and currency in nominal amounts were taken from Díaz et al., (2016) for the period between 1938–2010 [15]. This series was completed (period 2011–2019) with another one extracted from the statistical bases of the Chilean Central Bank (2023) for the period 1986–2019 [19]. Again, the splicing of the series was performed by linear interpolation. All data are available in the S1 Data).

The SFD indicators were calculated based on the Eqs (2 and 3):

$$SFDfd_t = B.fis_t \; x \; Inv.CIM_t \qquad (2)$$

$$SFDpd_t = Y/G_t \; x \; Inv.CIM_t \qquad (3)$$

Where *Inv.CIM* represents the inverse of the *CIM*, being therefore a measure of institutional mistrust, and "t" stands for time (in years). Higher values of *SFD* are indicative of an increase in the levels of fiscal stress and a lower capacity of the State to deal with the increase conflict in society [2, 3].

**MMP.** The population mobilization potential (*MMP*) is an indicator of the existing tension in the common population stratum [2, 3]. It is the result of the product of two demographic components, proportion of the population residing in urban areas (*Nurb*) and proportion of the population aged 20–29 years (*A20-29*), and the inverse of the relative median income (*w*). Following Turchin's (2013, 2016) formulation [2, 3], the *MMP* variable is defined as (Eq (4)):

$$MMP = w_t^{-1} \times Nurb_t \times A[20-29]_t \qquad (4)$$

Where *w* is estimated from the ratio: median nominal income / nominal GDP per capita (*GDPpc*). The value ($w^{-1}$) is an indicator of the impoverishment of the population's income (social welfare inverse). The relative income (w) was calculated between 1938–1957 from the social income tables reconstructed by Rodríguez Weber for his historical estimates of the Gini index for Chile (1850–2009) [13], which consists of a matrix containing the number of people and average income by sector or occupation. The matrix is divided into two tables. The first one covers between 1860–1930 and is divided into 49 categories of income earners: 9 in agriculture (7 categories of landowners and two of workers), 3 in mining, 10 in industry, 2 in transportation, 20 in the State, 1 professionals and 1 servants. The second table covers the interval 1929–1970 (we used until 1957), and is made up of 116 categories. In this case, the income recipients of the various economic sectors are grouped into 4 categories: employers or employers, own-account workers, employees and workers (unpublished material: personal communication). For the period 1958–2019 we employ "Occupation and Unemployment of Greater Santiago" surveys [20], from the Microdata center of the Faculty of Economics and

Business of the University of Chile. The data corresponding to the month of June were taken and annualized. In the case of data for the years 85,98,00,07 and 09 (not reported), the values of "$w$" were calculated by linear interpolation. GDPpc was estimated based on nominal GDP and population data from the World Bank and Díaz et al., (2016) [15, 21].

The information corresponding to the proportion of the youth population (20–29 years old) between 1938–1950 was estimated by linear interpolation of the data collected from the historical National Censuses [22]. For the rest of the time series, the proportion of the youth population was taken from the report of the National Institute of Statistics (INE) entitled "Chile: Projections and Estimates of the Population 1950–2050" [23]. This document presents the projections by five years. In order to generate a series with annual data, the intermediate values between each five years were estimated by linear interpolation. Finally, urban population data was obtained directly from the World Bank and Diaz et al., 2016 [15, 21]. All data are available in the S1 Data).

**EMP.** The intra-elite tension indicator (*EMP*) is proportional to the product of the relative number of the population that makes up this group (*e*) and the inverse of the relative income per capita for people belonging to this social stratum (epsilon; ε) [2, 3] (Eq (5)).

$$EMP_t = e_t \times \varepsilon_t^{-1} \tag{5}$$

To estimate the relative size of the elite (e), the social tables reconstructed by Rodríguez Weber (1938–1957) [13] and the Occupation and Unemployment surveys of Greater Santiago (1958–2019) from the Microdata center of the Faculty of Economics and Business of the University of Chile were used [20]. In this case, the average annual income of the top 5% was taken as the recruitment criterion, considering that subpopulation with income above that threshold value as members of the elite. To estimate the proportion of the elite, instead of taking the value of the total population of the country, the size of the sample was considered. Throughout the entire period, the elite proportion estimated through this methodology ranged between 1.3 and 1.9%, with an average of 1.5%. These values seem to show that the criteria used are restrictive enough to be considered a good proxy.

The inverse of the relative income per capita for people belonging to elite (ε, epsilon) was calculated using the Turchin's Eq [6]

$$\varepsilon_t = \left(1 - w_t \lambda_y\right)/e_t \tag{6}$$

Where λ represents the relative size of the labor force (L/N). The labor force data (L) for the period (1938–2010) were taken from Díaz et al., (2016) [15], while for the 2011–2018 period values were used from the report "Indicators of underutilization of the labor forces work in Chile: Evidence from the national employment survey" [24]. The 2019 value was extrapolated linearly.

**PSI indicator.** The Political Stress Index (PSI) was calculated as follows [2, 3] (Eq (7)):

$$PSI = SFDfd_t \times MMP_t \times EMP_t \tag{7}$$

Alternatively, we use the Ortmans et al. (2017) [4] formulation to improve the fit (Eq (8)).

$$PSI = SFDfd_t^a \times MMP_t^b \times EMP_t^c \tag{8}$$

## Statistical analysis

We use simple and multiple linear regression analyzes in R program [25], with a confidence level of 95%, to describe the main trends in the relationships of the variables w, Nurb, A20-29, and e. To evaluate the distribution of the residuals, the Shapiro-Wilk test of the "olsrr" package was used [26]. The construction of the time series was carried out through the use of the "tseries" package [27]. To avoid spurious correlations, seasonality and trend were removed from the time series through differentiation and the application of logarithm. Subsequently, we employ multiple linear regression models with ARIMA errors, using generalized least squares ("gls") techniques, to analyzes relation between the different variables and their interactions. In this procedure, the temporal correlation present in the model errors was taken into account, which was evaluated through the use of ARIMA models. In this case, the "forecast" and "astsa" packages from R core were used [28, 29]. In the case of the series e we use Chow Test to determine trend changes, using the "strucchange"package of R [30]. To evaluate the correspondence between the political stress indicator (PSI.) and the observed political instability index, we used a nonlinear regression model ("nls") and estimated a pseudo-R2 to weight the fit, using the package "nls. multistart" [31]. In order to check the predictive capacity of the PSI* model on the observed political instability index (1938–2019), we used the cross-validation technique, using a subset of data from the original series of the PSI. model to build a new series predicted using the projection of a step forward (k-1) progressively, using the functions of the "tseries" package [27]. To corroborate the fit between the predicted series and the observed political instability data, the root mean square error (RMSE) was estimated. Finally, for the graphic representation, all the variables were standardized on a 1–2 scale. In order to show the central trajectory of the variables, moving averages (time window of 5 years) and the loess function were used. We use "ggplot2" package to elaborate all graphs [32]. For the statistical analyses, series with moving averages (5-year window) were used.

## Results

### Dynamics of the three structural variables (MMP, EMP and SFD)

The dynamics of the three structural variables (MMP, EMP and SFD) exhibits a fluctuating pattern during the last 80 years (Fig 1). Fig 1A shows the trajectory of the two indicators used to estimate the SFD variable, in red the index based on relative external debt and in black the one based on fiscal stress (Fig 1A). As a general trend, three major periods of stress accumulation are observed in both indicators: from 1938 to the mid-1950s (more conspicuous in the black curve), from the early 1960s to the mid-1970s and from the early 1980s to the mid-1980s (more conspicuous in the red curve). Finally, starting in the late 1980s onwards, an irregular decline is observed in both indicators until the beginning of the 2010s, when they showed a slight increase. (Fig 1A). The MMP variable shows a slight increase in its values from the beginning of the series until the second five years of the 1960s, when the stress levels predicted by this indicator register a strong acceleration. After this period, the MMP values follow an increasing trend, which is mainly noticeable between the end of the 1990s and the beginning of the 2010s (Fig 1B). In the case of the EMP trajectory, our results estimate two long periods of accumulation of intra-elite tension: from 1938 until the mid-1970s with an increase in acceleration from the late 1950s, and from the end of the 1980s to the present (Fig 1C). The variation over time of each of the components of these three structural variables is shown in the supplementary materials (S2 Fig).

Fig 1D shows that the periods in which an upward trajectory of the PSI indicator is obtained, which according to SDT is indicative of an increase in political stress, coincide

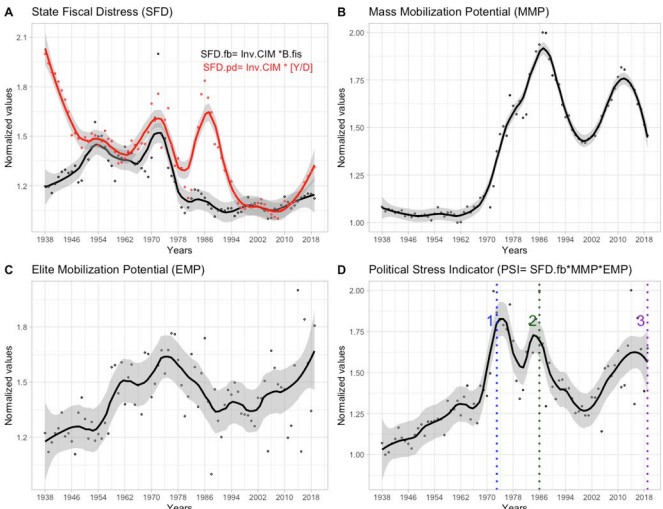

**Fig 1. Dynamics of the structural variables 1938–2019 (EMP, MMP, SFD) and trajectory of the political stress indicator predicted by the structural-demographic theory (PSI).** A) Fiscal stress indices (SFD.fb [black line] & SFD. pd [red line]). B) Population Mobilization Potential (MMP). C) Intra-elite tension (EMP). D) Political Stress Index (PSI). Numbers are representative of the main events of conflict: 1- Coup d'état of 73 (blue), 2- Popular mobilizations of the 80s (green), 3- Social Outbreak 2019 (violet). *Smooth with loess method (span = 0.25). Shaded area represents confidence interval (95%). All data were normalized to a scale between 1–2.*

temporally with the occurrence of major events of social instability in the last 80 year: 1-State coup of 1973, 2- Cycle of popular protests during the 1980s, 3- "Social outbreak" of 2019. Regarding the period prior to the 1973 coup, it is observed that there is a growth concomitant in the accumulated stress in the 3 structural variables (SFD, MMP & EMP) (Fig 1A–1C). In the case of the massive protests of the 1980s, they occur in a context where the SDT suggests an increase in the accumulation of tension in the MMP and SFD variables (Fig 1A and 1B). Finally, the events related to the "Social Outbreak" overlap with increases in the MMP and EMP values, although it coincides with a SFD that remains stable (Fig 1A–1C).

## Correspondence between observed political instability index and political stress indicator (PSI) (1938–2019)

Fig 2A shows the results obtained by comparing the dynamics of the PSI index and the observed political instability indicator. The observed dynamics of the political-institutional instability experienced by Chile in the period 1938–2019 were reconstructed using a detailed review of the literature (see materials and methods and supplementary materials). Within our period of interest, we were able to identify three historical moments where conflict events are concentrated. The first peak corresponds to the period between 1967 and the mid-1970s. In this period, the most important event corresponds to the coup d'état of September 1973. A second peak in the dynamics of political tension is observed in the period between the early and mid-1980s). Finally, the third recorded peak occurs at the end of the time series and indicates a marked increase in conflict values dating from the early 2010s (Fig 2A).

The non-linear regression analysis (NLS) indicates the existence of a statistically significant relationship between the PSI* indicator and the observed political instability dynamics (pseudo-$R^2$ = 0.55; parameter values: a = 0.54, p-value = 1.86e-11*; b = 0.36, p-value = 2.60e-08*; c = 0.07, p-value = 0.421). Alternatively, all possible combinations derived from the main model (PSI* = SFD $^a$ x MMP $^b$ X EMP $^c$) were evaluated, considering the product of 2 components and each of the individual components separately (S1 Table). The results obtained show

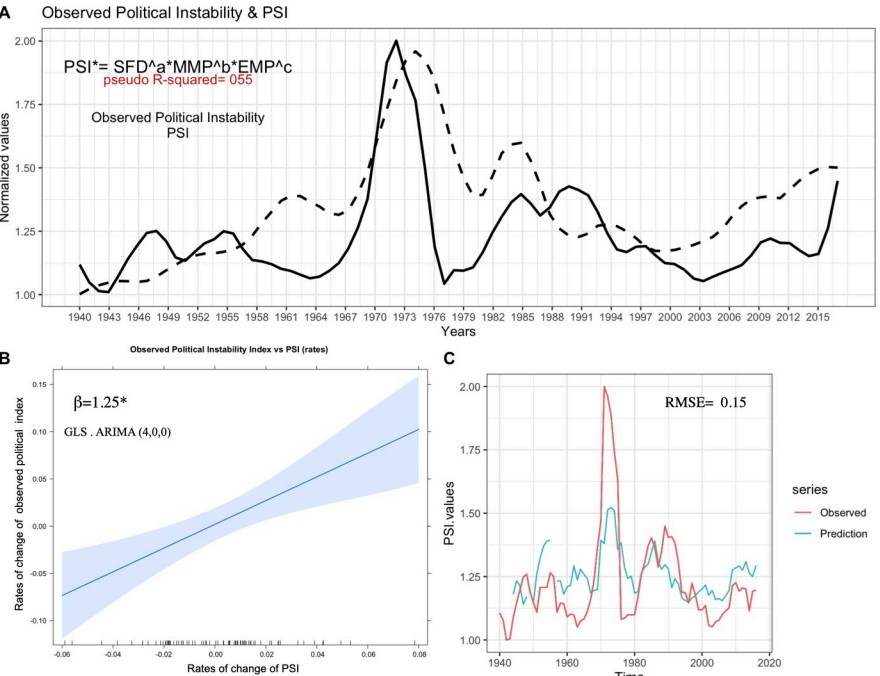

**Fig 2. Correspondence between the observed political instability index and the political stress indicator predicted by the structural-demographic theory (PSI) (1938–2019).** A) Comparison of the dynamics of the PSI indicator (firm line) and the observed political instability index (dotted line). *Regression between PSI\* (PSI\* = $SFD^a*MMP^b*EMP^c$ ) and observed political instability index in Chile (1938–2019). NLS Parameter values: a = 0.54 (p-value = 1.86e-11\*), b = 0.36 (p-value = 2.60e-08\*), c = 0.07 (p-value = 0.421), pseudo-$R^2$ = 0.41. Residual: Test Shapiro-Wilk. 0.913, p-value = 0.48* B) GLS Regression between rate of change of PSI\* (PSI\* = $SFD^a*MMP^b*EMP^c$) and rate of change of observed political instability index in Chile (1938–2019). *(ARIMA (4,0,0), β = 1.25. CI-95%: 0.57–1.93, p-value = 0.0005\*).*C) Prediction through the Cross-Validation method (prediction one step ahead) of the dynamics of political instability in Chile from the PSI\* model (1938–2019). *Predicted series (green) and Observed series (red). RMSE = 0.15. All data were normalized to a scale between 1–2.*

that the best model in terms of fit and complexity is PSI\* (pseudo-R2 = 0.55, AIC = -67.9). However, it is important to highlight that the alternative model $SFD^a$ x $MMP^b$ also presents an important relationship with the observed political instability dynamics (pseudo-R2 = 0.53, AIC = -69.2).

The regression with ARIMA errors model between the rate of change of PSI\* (see methodology) and the rate of change of the observed political violence index shows a positive linear relationship between the two indicators (β = 1.25. CI-95%: 0.57–1.93, p-value = 0.0005) (Fig 2B). On the other hand, the predictions obtained when using the PSI\* model, by the cross-validation technique and the projection one step forward (k-1), are shown in Fig 4C. It shows an important correspondence between the dynamics of political instability observed and that predicted through the SDT model (RMSE = 0.15) (Fig 2C). Even, this strong correspondence is maintained by increasing the prediction window of the model several steps forward (2,4,6 and 8 steps). In this case, although the prediction noise increases, the dispersion levels of the residual values (RMSE) still show a good fit between the political instability predicted by the model and the observed political instability index (S3 Fig). These results seem to indicate that structural demographic theory (SFT) effectively predicts the main periods of increased social tension and political instability in Chilean society over the last 80 years.

## Relationship between population well-being (w) and relative number elite overproduction (e)

Fig 3A shows the trajectories of the median relative income of the population (w; proxy of general well-being) and the relative number of people who belong (or aspire to belong) to the elites (e). It shows that, in general terms, both variables present an antithetical trend, where periods of growth of "e" seem to overlap with scenarios of decrease of "w" (and vice versa). This tendency in the dynamics of both variables seems more evident between the periods: 1938 and the beginning of the 80s and from the late 90s to the end of the series (Fig 3A). The multiple regression analyzes shown in Fig 3B seem to corroborate the existence of a certain negative relationship between "w" and "e" and their interaction with demographic variables: proportion of young population [(A20-29)/Youth] ($R^2 = 0.48$, $\beta1 = -0.81$, $\beta2 = -0.28$, p-value = 1.11e-11*) and proportion of urban population [Nurb]) ($R^2 = 0.83$, $\beta1 = -0.14$, $\beta2 = -0.80$, p-value = <2.2 e-16*).

To corroborate the interactions described above, regressions (gls) were performed considering autoregressive errors employing ARIMA models, using the rates of change of the components of the structural variables MMP and EMP. The results obtained support the existence of a negative relationship between the rate of change of the relative income of the population (w) and the rate of change of the relative number of elites (e) in interaction with the rates of change of the demographic variables: proportion of youth population ([A20-29]; Youth) (ARIMA (2,0,2); $\beta = -5.62$, p-value = 0.001*) (Fig 4A) and proportion of urban population ([Nurb]) ($\beta = -20.02$, p-value = 0.04*) (Fig 4B). In other words, the increase in the relative number of elites (positive values) is negatively associated with the rate of change of relative income when there are increases in the proportion of youth population and in the proportion of urban population (Fig 4A and 4B), suggesting that the negative relationship between the rate of change of "w" and rate of change of "e" is mediated by the changes in these two demographic variables, (i.e., *A20-29* and *Nurb*).

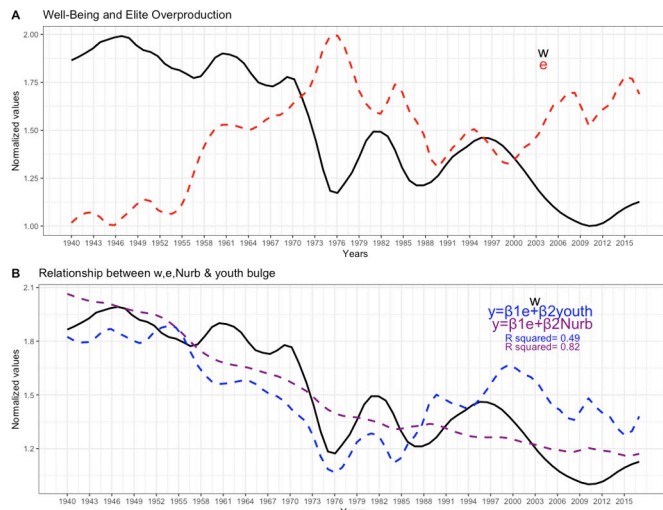

**Fig 3. Analysis of the interactions of the EMP and MMP components.** A) Antithetic trend between the relative income of the population (w; black) and the relative number of people that make up the elites (e, red). B) Relationship between demographic variables and the relative income of the population. *Multiple linear regression between w (black series), and e & proportion of youth population (blue series) ($R^2 = 0.48$; $\beta_1 = -0.82$; $\beta_2 = -0.28$; p-value< 2.81 e-12*). Multiple linear regression between w, and e & proportion of urban population (violet series) ($R^2 = 0.82$; $\beta_1 = -0.11$; $\beta_2 = -0.80$; p-value< 2.2 e-16*).*

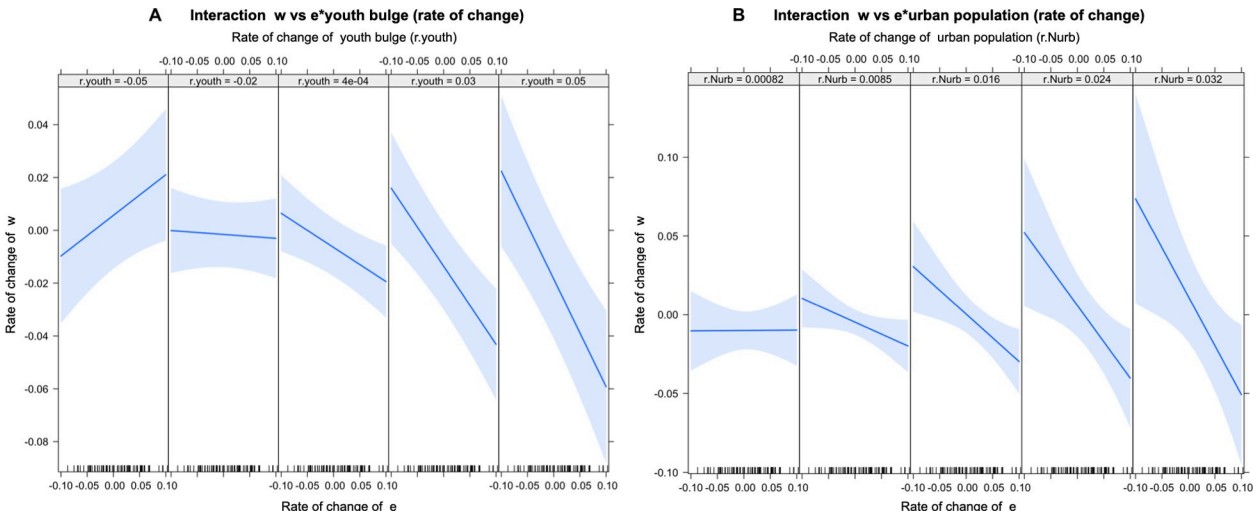

**Fig 4. Relationship between rate of change of w (*rw*) and rate of change of e (*re*) mediated by demographic variables.** A) Relationship between rate of change of w (*r.w*) and the interaction of rate of change of e (*r.e*) and youth proportion (*r.youth*): (*ARIMA* (2,0,2); *β* = −5.62. *C.I-95%* = (-8.35,-2.88), *p-value* = 1e-3*. *Residual*: *Shapiro-Wilk test* = 0.9291, *p-value* = 3e-4). B) Relationship between *r.w* and the interaction of *r.e* and *r. Nurb*. (*β* = −20.02; *C.I-95%* = (-38.8,-1.23), *p-value* = 0.04*. *Residual*: *Shapiro-Wilk test* = 0.8485, *p-value<0.01*). Shaded area represents confidence interval (95%). All data were normalized to a scale between 1–2.

## Discussion

Structural demographic theory (SDT) is an interpretative framework to evaluate the extent to which the dynamics of structural variables, related to the resource requirement/access ratio in different social actors and fiscal stress, can predict the trajectory of political tension in a given social system [1, 2]. This approach does not pretend to cover all the complexity associated with the emergence of major events of socio-political instability, nor does it ignore the importance of the agency of political actors in these processes. Only, based on a simple model, it seeks to delineate the socio-economic contexts in which the different social agents are more likely to mobilize and initiate political conflict.

Our results show that in the case of Chilean society over the last 80 years, major periods of political conflict were preceded by scenarios in which the growth of certain social groups exceeded their accessibility to resources. These processes tend to increase internal competition, affecting the relationship between elites, popular groups, and the State, impacting the stability of the institutional system [33]. In this sense, the rising dynamics of the PSI political stress indicator seem to represent the periods in which the Chilean institutional system is more vulnerable to experiencing the emergence of major political conflict events.

Although the results presented are consistent with the predictions derived from the adaptation of SDT to the Chilean case, they do not empirically establish causality. The outbreak and outcome of a political instability event depend on multiple factors that are beyond the theoretical framework of SDT and require other types of approaches that emphasize organizational, ideological, and strategic aspects (i.e., the existence of charismatic leaders, ideological identification processes, international pressures, organizational capacity, and linkage of elites with popular groups). Nevertheless, our results suggest that for the Chilean case, the SDT model provides good probabilistic estimates of the stability of the institutional system.

Despite its relative simplicity, the political stress index (PSI) predicted by the structural-demographic model captured the occurrence of social conflict and political vulnerability by predicting levels of discontent in different social groups. The model is well-adjusted in spite of

the profound changes experienced in political and economic institutions in the last 80 years [1, 2]. This adjustment capacity is sustained even when using multi-step forward predictive methodologies (up to 8 years). In this sense, our results indicate that, at least within the period of interest, monitoring the dynamics of the 3 structural variables allows us to anticipate the state of stability of the institutional system. Future studies could advance conceptual frameworks that allow the integration of SDT with other approaches, which would contribute to a better understanding of the complexity of these social phenomena and how the vulnerability predicted by structural demographic theory combines with other factors to open the opportunity for the mobilization of different social groups.

Our results show that the SDT predicts an increase in political tension in Chilean society prior to the emergence of the main social instability events of the last 80 years. By analyzing the qualitative dynamics of each of the 3 components of the PSI indicator (SFD, MMP, and EMP), it is possible to recognize some general trends. All events of political instability are preceded by an increase in tension in the popular sectors. This observation is corroborated in part by the high fitted values of the alternative model (SFD[a] x MMP[b]) when compared to the trajectory of the observed dynamics of the political instability index. At the same time, prior to the coup d'état (1973) and the cycle of mobilizations in the 1980s, significant increases were observed in one of the two indicators used as a proxy for SFD, as well as a more moderate increase from the beginning of the 2010s. Regarding the elites, tension in this group increased prior to the coup of 1973 and the social outburst of 2019, while in the 1980s it showed a downward trajectory.

In this sense, according to our estimates, the increase in tension among the elites precedes the events of political conflict that led to profound changes in the institutional regime (the coup d'état of 1973) or in processes of opening to potential reforms (the Social Outburst of 2019). On the other hand, in periods of accumulation of fiscal stress and increased pressure on the general population without accompanying of elite mobilization potential, although scenarios of social conflict emerged (mobilizations of the 1980s), they did not lead to institutional changes. Therefore, the above observations could suggest the potential existence of a correspondence between the dynamics of structural variables and certain characteristics of the degree of magnitude and probability of success of emerging conflict events. With the information available to date, it is too early to draw a conclusion in this regard. Future studies focused on the reconstruction of the 3 structural variables in different contemporary societies will allow us to go deeper into these aspects. Regarding the 2019 "Social Outburst", our model predicted it with reasonable accuracy, suggesting a scenario of social mobilization accompanied by intensified competition between the working class and the elite and intra-elite competition. Since the second half of the 1990s, the rate of growth of the segment of the Chilean population that makes up or aspires to make up the elite exceeds the capacity of the system to sustain it. Such growth amplified competition and helped generate political destabilization. At the same time, the relative income of the population (w) shows an irregular declining trend since the beginning of the 1970s. Therefore, the structural-demographic theory predicts a pre-2019 context of intense intra-elite competition and significant discomfort among the population due to relative deprivation, which overlaps with maximum levels of institutional disaffection (S4 Fig). The conjunction of these factors constituted an explosive scenario.

Our results thus suggest that elite overproduction (characterized by the growth of the EMP values) is one of the main drivers of the dynamics of political instability in Chile during the period 1938–2019, especially of the conflicts that led to the most significant institutional changes, as in 1973 and 2019. Our analysis also suggests diminishing returns to the growth rate of the elite given limited resources. In this sense, social processes that increase the number of people with credentials that lead them to aspire to elite status will tend to generate intra-

group tension because the surpluses generated by the socioeconomic system remain relatively inelastic. This factor can be especially pressing in Chile, which has a "winner take all" economy in which the highest-earning 1% of the population captures close to 33% of the national income [8, 34, 35]. This characteristic place a substantial limit on the availability of resources to meet the requirements of the new aspiring elite, which translates into a greater intensity of competition and increased division among the elite, creating opportunities for the emergence of political instability.

Consistently, since the 2000s, there has been a significant increase in the frequency of events associated with the polarization of the elite, such as public denunciations of corruption scandals, breakdown of political consensus, conflicts between congressional and presidential powers, cabinet secretary turnover, and constitutional indictments. At the same time, a growing scenario of political fragmentation has also been observed (S5 Fig). These trends seem to correspond to the intensified competition dynamics predicted by structural-demographic theory for elites, a situation associated with relative suppression, radicalization, and intra-group alienation [36].

The increase in the number of people who aspire to occupy elite spaces in the last 20–25 years accounts for the growth of groups associated with financial management, business and the ownership of companies related to extractive and commercial activities. As a result of Chile's greater participation in international markets, these sectors have grown substantially during the second half of the1990s [13, 37, 38]. On the other hand, starting in the mid-2000s, with the introduction of a state-endorsed credit system (CAE), there has been a substantial increase in enrollment in tertiary education, as well as a marked increase in the number of professional graduates in the country [39–41]. The socioeconomic system could not absorb the growth in the number of people with sufficient educational credentials to aspire to social mobility and potential positions of power [39–41].

On the other hand, along with the process of overproduction of the elites, in recent decades there has also been an increase in the levels of MMP as a consequence of the dynamics of relative income (w). In this way, in a context of massive liberalization of the economy and a deepening of the globalization of value chains, as of 1998 a significant drop in the values of "w" is observed. Both factors have prevented the recovery of workers' bargaining power, which has kept the values of "w" far from those estimated at the beginning of the series [10, 13]. In this sense, our estimates suggest that higher values of "w" correspond to periods of greater bargaining power on the part of workers' organizations and the orientation of the economy towards the domestic market.

Our results show that for the Chilean case the increase in the number of people who aspire to occupy positions of power in society can also have repercussions on the relative income values of the population (w). In line with Turchin's statement [2–4], the latter could be a consequence of the fact that the increase in intra-elite competition generates incentives in the groups that make up this stratum to seek to capture a greater portion of the resources, which reduces the "share of the pie" that is captured by the general population [1, 2]. In our results, the negative pattern between the rates of change of "e" and "w" is reinforced by the growth of the urban population and the youth population. In this sense, the expanding elite, together with an accelerated increase in the labor supply (due to the growth of the urban population and of young people), will tend to generate the conditions for the relative income of the population to experience pressures to the low, decreasing or slowing its growth. According to our estimates, this trend in Chile is independent of the changes observed in economic institutions between 1938–2019, a period in which the country experienced a state-led industrialization model (∼1940–1973) and a job allocation model resources centered on the free market (from 1973) [13]. Therefore, our results suggest that the overproduction of elites constitutes one of the most potentially destabilizing processes in the Chilean social system in the last 80 years,

since in addition to increasing intra-elite tension, it increases the potential for population mobilization by putting downward pressure on relative income (w) in contexts of demographic growth.

In sum, Chilean political stability seems to be pressured by two joint forces, namely, the overproduction of elites combined with a decrease or stagnation of the relative income of the population. This context creates a powder keg where apparently minor episodes can trigger events that cause great sociopolitical instability.

Politicians and political pundits often interpret events like Chile's "Social Outburst" as the result of contingent agency by specific groups and their leadership. For instance, in 2019, the Chilean government attributed the riots to a plan for political destabilization perpetrated by extreme leftist groups and their international allies. Our results suggest, instead, that the events of 2019 correspond to a "collective action cascade" triggered by contingent events but enabled by an underlying set of structural factors [42]. Given that Chile is often considered a development model for other Latin American countries, our results further point to possible structural weaknesses of such a predicament. On this note, future works might seek to extend the application of the SDT to other instances of political instability in the region to test the external validity of our findings.

We also found the increasing levels of political conflict observed in Chile during the last decade are synchronic with the higher levels of political tension in much of Latin America (S6 Fig). Thus, the current great integration of markets at a global level allows the expansion of an exogenous disturbance in broad geographic areas, which could contribute to a rapid alignment of stable endogenous dynamics, resulting in a synchronization of the cycles. The globalization of the economy involves enterprises transferring significant parts of their production to those countries where labor costs are lower, increasing their control over financial activities and bypassing the regulatory mechanisms of the nation-states [43].

Another consequence of the globalization process is a constant reduction in relative wages in the Western world, explained by the greater bargaining power of capital during diminishing returns and amplifying the tensions among the elite and the general population. Unfortunately, this is not a trend for which we can foresee any significant changes in the near-term [43]. A crucial implication of our study is that policy-makers and pro-democratic forces interested in the long-term prospects of democracy should thus devise a new "social pact" that can better counteract the structural forces (growing inequality; an overproduction of aspiring elites) that induce democratic crises in contemporary liberal regimes. Such "social pact" should include traditional social policy targets such as health, education, employment, and social security and innovative ways of tackling inequality and the social mobility expectations of new generations. Moreover, the "social pact" should be compatible with a new era in which pursuing ecologically sustainable economic growth is inescapable.

## Supporting information

**S1 Fig. Dynamic of the observed political instability indicator.**
(TIF)

**S2 Fig. Dynamics of the components of the structural variables SFD, MMP and EMP (1938–2019).** A) SFD components: Budgetary income and expenses (B.fis, black), ratio between public debt and gross domestic product (Y/G, red dotted series) and inverse of intensive contract money index (Inv.CIM, blue dotted series). B) MMP components: Median relative income of the population (w, black series), relative urban population (Nurb, blue series) and proportion of population aged 20–29 ([A20-29], red series). C) EMP components: Relative number of people who belong to or aspire to elite status (e, black series) and relative per capita

income of elites (epsilon, red dotted series). Smooth with loess method (span = 0.3). All data were normalized to a scale between 1–2.
(TIF)

**S3 Fig. Prediction through the cross-validation method of the dynamics of political instability in Chile from PSI\* model (1938–2019).** A) Two steps forward (h = 2) (RMSE = 0.152). B) Four steps forward (h = 4) (RMSE = 0.161). C) Six steps forward (h = 6) (RMSE = 0.172). D) Eight steps forward (h = 8) (RMSE = 0.184). Predicted series (RED) and Observed series (black). All data were normalized to a scale between 1–2.
(TIF)

**S4 Fig. Institutional skepticism proxies.** Expectations regarding the future situation of the country (stagnation/decay) (red), government disapproval (green), no identification with political parties (blue) and no identification with the left-right axis (purple). Authors' elaboration based on data extracted from the CEP survey (1990–2018) Smooth with loess method (span = 0.3). Shaded area represents confidence interval (95%). All data were normalized to a scale between 1–2.
(TIF)

**S5 Fig. Intra-elite competition indicators.** A) Dynamics of the Simpson Index (1989–2021). Probability of selecting two deputies in the lower house and that both are from the same political party. B) Dynamics of the Simpson Index (1989–2021). Probability of selecting two deputies in the lower house and that both are from the same political alliance. C) Number of impeachments initiated in the parliament.
(TIF)

**S6 Fig. Dynamics of political conflict in Latin America.** Weighted annual indicator: anti-government demonstrations (\*10) + riots (\*25). Data taken from Banks et al. (2021). Smooth with loess method (span = 0.3). Shaded area represents confidence interval (95%).
(TIF)

**S1 Table. Alternative models.** Analysis of the relationship with the observed political instability index.
(DOCX)

**S1 Data.**
(XLSX)

**S2 Data.**
(R)

## Acknowledgments

We thank Javier Rodríguez Weber for his valuable contribution in providing essential data for the development of this research.

## Author Contributions

**Conceptualization:** Manuel Muñoz-Rodríguez, Juan Pablo Luna, Mauricio Lima.

**Data curation:** Manuel Muñoz-Rodríguez.

**Formal analysis:** Rosana Ferrero.

**Funding acquisition:** Mauricio Lima.

**Investigation:** Manuel Muñoz-Rodríguez.

**Methodology:** Manuel Muñoz-Rodríguez, Rosana Ferrero.

**Supervision:** Juan Pablo Luna, Mauricio Lima.

**Writing – original draft:** Manuel Muñoz-Rodríguez, Mauricio Lima.

**Writing – review & editing:** Juan Pablo Luna, Mauricio Lima.

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
