## [Decision Letter · Decision Letter 0]

10 Jul 2023

PONE-D-23-15713Squeezed from above: Sociopolitical dynamics in Chile (1938-2019) and its relationship with the process of overproduction of the elites.PLOS ONE

Dear Dr. Muñoz-Rodríguez,

Thank you for submitting your manuscript to PLOS ONE. After careful consideration, we feel that it has merit but does not fully meet PLOS ONE’s publication criteria as it currently stands. Therefore, we invite you to submit a revised version of the manuscript that addresses the points raised during the review process.

First evaluator

This paper presents an interesting application of Structural Demographic Theory (SDT) to the case of Chile in the last century. It mostly draws on Goldstone and Turchin’s work, but it seems to go a bit further in terms of methodology by incorporating predictive analytics. The model used achieves to predict social unrest periods, leading the authors to argue that it shows the “underlying cause of Chilean political instability”. Although I think this is a very interesting piece of work, I believe it would benefit from some changes that I would suggest hereafter:

First, I will address some form issues.

I recommend modifying the title of the article to explicitly indicate that it is an application of SDT. That way it would be easier to identify the paper as a part of a broader literature, and would make it more accessible to a wider audience beyond those specifically interested in the Chilean case. Second, I suggest moving the materials and methods section before Results. Such order is common among PLOS articles and it would make a lot easier to understand the results section. Lastly, it appears that there are some issues with the equations when exporting from the text editor (see lines 73 and 348) and some equations seem to have errors (I’m guessing in eq 1 frequency and total occurrence are indexed in i).

Introduction:

This section seems very clear to me. It frames adequately SDT as an “interpretative framework” that “does not seek to encompass all the complexity” behind political instability but is sufficient to model (and predict) social unrest periods. In terms of the explanation of the Chilean case, it is weird that it goes into detail about social unrest during 2019, but does not address other important political instability events during the time window studied. It seems important to provide contextual information on political instability throughout the 20th century in Chile, as the study later treats at least two other events with the same level of importance as the "social outbreak" of 2019.

Results:

Once the goal of the study is introduced, the paper jumps to the results section with a visual inspection of four time series, one for each independent variable and one for an index that aggregates them with a multiplicative model. The visual inspection closes with a big conclusion: “it can be seen how the structural demographic theory predicts an increase in political tension in the Chilean social system prior to the outbreak of the main events of social instability of the last 80 years”. This is one of the main reasons why I recommend putting the methods section before. As I read this without knowing the methods that were going to be used to test predictability, it was easy to conclude the study was flawed. Visual inspection does not support predictability. I suggest either removing this initial part of the results or moderating the conclusion.

Then, the outcome of the model is presented. The authors introduce the Observed Political Instability Index (OPII) and while it can be inferred that this measure represents the dependent variable created by the authors, it would be a lot clearer if the methods section were before the results.

Finally, prediction models are presented: a non-linear regression proves that SDT can fit OPII, Arima model shows a temporal correlation between changes in observed and estimated outcomes, and forecasting provides evidence that SDT can predict the OPII using information from a year before. Unfortunately, this core aspect of the study, which addresses the research question of whether SDT can predict social instability, is described in a brief paragraph. My suggestion is to extend the explanation of these results and go deeper into them. You could go further in the explanation of the magnitude of the parameters, test if the prediction would stand for a bigger time horizon, or identify which structural variable is more important to the prediction (which appears to be an important aspect in the Discussion section).

The last section of the results shows the interaction of two of the dimensions of SDT. This is coherent with the theory that stays that there are feedback loops between the variables. It is not clear why this was the only interaction tested. I suggest stating in the introduction a clear research question that is answered by this analysis, explaining in the results section why it is the only interaction addressed, or extending this section to include tests for other interactions.

As a final comment for this section, I would like to emphasize that these results discuss predictability but do not establish causality. I strongly recommend being cautious in making any claims that imply causation has been proven.

Discussion:

The discussion gives a rich explanation of the Chilean case and why it is in line with the results of the study. Although I think this is very valuable, it would be important to mention something about the policy implications of your results, taking into account that in the introduction you stated “Understanding the structural forces that underlie the dynamics of political instability in modern societies is essential to adopt measures that allow adapting social systems prior to the irruption of events” What can a policy maker learn from your study? I would also like to see some discussion about the implication of the results to SDT.

I find the mention of social distress in other countries highly relevant, considering that these countries likely exhibit significant variations in structural variables. It is of course beyond the scope of this study, but it would be interesting to see comparative studies of SDT.

Methods:

I cannot stress enough the recommendation of putting this section before results. Besides that, I suggest declaring more clearly that not only the model of SDT but also the measures are based on Turchin’s previous work. That would limit questions to the construction of structural variables and indexes.

This section should also explain the sources used for every measure and include more information that clarifies and justifies the chosen methods. Most of this is already written in supporting information, but it should be explained to some extent in the body of the paper.

Two last important points. The data provided do not meet PLOS politics. The data points used to create the provided aggregated measures, as well as the code used to process the information, should be included. Also, the financial disclosure link is broken.

Second evaluator

The paper presents a novel methodological and theoretical proposal for the analysis of socio-political dynamics in the Chilean context. The structural demographic theory is introduced and applied to approach the process of overproduction of the Chilean government elites and its relationship with the processes of mobilization and social conjuncture that the country has experienced. This aspect is in fact an element to be highlighted since it allows us to approach an understanding of the complex phenomena of social mobilizations and contributes to the development of future research in the región.

The following are a series of formal comments to be taken into consideration by the authors:

1) While it is true that at the introductory level the authors address the Structural Demographic Theory and the way it will be applied in the study, I suggest expanding a little more in detail on the scope of this theory to the study of social dynamics. Social demographic theory provides a conceptual framework for understanding patterns of population change at different stages of a country's socioeconomic development. The authors have supported with an exhaustive level of detail the methodological guidelines to account for the dynamics of change in the Chilean context with the variables for this purpose. However, the reader may have the feeling that the complex framework that Chile has experienced in this period (1938-2019) is reduced to a brief analysis of variables, omitting the complexity of the political, social, cultural, and historical reality of the country.

2) The phenomenon of social mobilizations is in essence a complex phenomenon and obeys the particularity of a specific context and time. Several elements have been mentioned that accompany the emergence of mobilization. While it is true that the authors mention in the introductory section of the manuscript, I quote: "Therefore, SDT does not seek to encompass all the complexity associated with the phenomenon of political conflict in social systems, nor does it ignore the importance of the agency of political actors. It only allows us to predict, through the interaction of structural variables, periods when the collective mood is more likely to trigger major scenarios of institutional instability" and in fact, the study does not seek to reduce the complexity of the social reality of Chile and the social mobilizations of October 18 to an analysis of variables, I suggest that the authors expand in more detail in the introductory section these elements of discussion.

3) The manuscript is structured from an introductory section followed by a middle section that introduces the reader to the findings of the study and then the reader will find a methodological section. I suggest that the methodological section comes first before the results. The manuscript is accompanied by supplementary material that the reader can consult to go into more detail on how the analysis proceeded. Nevertheless, the material contains important information that would be worth including in the methodological section of the article. To expand on the details of the procedure, the collection of the empirical corpus, the databases consulted, the previous works consulted. It would be more convenient for the reader to find this information in the methodological section and leave the more technical information guiding the methodological elaboration of the study in the supplementary file. Equations etc.

4) Finally, I suggest to the authors to expand the discussions of the study in terms of the scope of this research for the development of future studies in the region. How the use of Demographic Structural Theory can be a conceptual framework to address regional inequalities and disparities orchestrated by the elites. In terms of the challenges for the case of Chile to agitate planning processes and public policies for the future. By understanding the stages of transition that Chile has experienced in its history, it would be possible to infer changes in the population structure and think about the possibility of adapting policies in areas such as health, education, employment, social security, etc.

General considerations of the Academic Editor:

I suggest that the following recommendations of the evaluators be prioritized

a. Is the manuscript technically sound, and do the data support the conclusions? PARTLY

b. Has the statistical analysis been performed appropriately and rigorously? NO

c. Have the authors made all data underlying the findings in their manuscript fully available? NO

d. Change the order of the sections of the article (method before results)

We look forward to receiving your revised manuscript.

Kind regards,

Julio Cesar Ossa, Ph.D

Academic Editor

PLOS ONE

Journal Requirements:

"We thank the Center for Applied Ecology and Sustainability (CAPES), ANID PIA/BASAL FB0002 and the ANID Advanced Human Capital Program for funding this research. We also thank Javier Rodríguez Weber for his valuable contribution in providing essential data for the development of this research."

"Center for Applied Ecology and Sustainability (CAPES), ANID PIA/BASAL FB0002

MM received a grant: ANID Advanced Human Capital Program for funding this research (https://www.conicyt.cl/becasconicyt/2014/08/28/beca-doctorado-nacional-2015/)

3. We noted in your submission details that a portion of your manuscript may have been presented or published elsewhere. [NOThilo Gross] Please clarify whether this [conference proceeding or publication] was peer-reviewed and formally published. If this work was previously peer-reviewed and published, in the cover letter please provide the reason that this work does not constitute dual publication and should be included in the current manuscript.

Additional Editor Comments:

First evaluator

This paper presents an interesting application of Structural Demographic Theory (SDT) to the case of Chile in the last century. It mostly draws on Goldstone and Turchin’s work, but it seems to go a bit further in terms of methodology by incorporating predictive analytics. The model used achieves to predict social unrest periods, leading the authors to argue that it shows the “underlying cause of Chilean political instability”. Although I think this is a very interesting piece of work, I believe it would benefit from some changes that I would suggest hereafter:

First, I will address some form issues.

I recommend modifying the title of the article to explicitly indicate that it is an application of SDT. That way it would be easier to identify the paper as a part of a broader literature, and would make it more accessible to a wider audience beyond those specifically interested in the Chilean case. Second, I suggest moving the materials and methods section before Results. Such order is common among PLOS articles and it would make a lot easier to understand the results section. Lastly, it appears that there are some issues with the equations when exporting from the text editor (see lines 73 and 348) and some equations seem to have errors (I’m guessing in eq 1 frequency and total occurrence are indexed in i).

Introduction:

This section seems very clear to me. It frames adequately SDT as an “interpretative framework” that “does not seek to encompass all the complexity” behind political instability but is sufficient to model (and predict) social unrest periods. In terms of the explanation of the Chilean case, it is weird that it goes into detail about social unrest during 2019, but does not address other important political instability events during the time window studied. It seems important to provide contextual information on political instability throughout the 20th century in Chile, as the study later treats at least two other events with the same level of importance as the "social outbreak" of 2019.

Results:

Once the goal of the study is introduced, the paper jumps to the results section with a visual inspection of four time series, one for each independent variable and one for an index that aggregates them with a multiplicative model. The visual inspection closes with a big conclusion: “it can be seen how the structural demographic theory predicts an increase in political tension in the Chilean social system prior to the outbreak of the main events of social instability of the last 80 years”. This is one of the main reasons why I recommend putting the methods section before. As I read this without knowing the methods that were going to be used to test predictability, it was easy to conclude the study was flawed. Visual inspection does not support predictability. I suggest either removing this initial part of the results or moderating the conclusion.

Then, the outcome of the model is presented. The authors introduce the Observed Political Instability Index (OPII) and while it can be inferred that this measure represents the dependent variable created by the authors, it would be a lot clearer if the methods section were before the results.

Finally, prediction models are presented: a non-linear regression proves that SDT can fit OPII, Arima model shows a temporal correlation between changes in observed and estimated outcomes, and forecasting provides evidence that SDT can predict the OPII using information from a year before. Unfortunately, this core aspect of the study, which addresses the research question of whether SDT can predict social instability, is described in a brief paragraph. My suggestion is to extend the explanation of these results and go deeper into them. You could go further in the explanation of the magnitude of the parameters, test if the prediction would stand for a bigger time horizon, or identify which structural variable is more important to the prediction (which appears to be an important aspect in the Discussion section).

The last section of the results shows the interaction of two of the dimensions of SDT. This is coherent with the theory that stays that there are feedback loops between the variables. It is not clear why this was the only interaction tested. I suggest stating in the introduction a clear research question that is answered by this analysis, explaining in the results section why it is the only interaction addressed, or extending this section to include tests for other interactions.

As a final comment for this section, I would like to emphasize that these results discuss predictability but do not establish causality. I strongly recommend being cautious in making any claims that imply causation has been proven.

Discussion:

The discussion gives a rich explanation of the Chilean case and why it is in line with the results of the study. Although I think this is very valuable, it would be important to mention something about the policy implications of your results, taking into account that in the introduction you stated “Understanding the structural forces that underlie the dynamics of political instability in modern societies is essential to adopt measures that allow adapting social systems prior to the irruption of events” What can a policy maker learn from your study? I would also like to see some discussion about the implication of the results to SDT.

I find the mention of social distress in other countries highly relevant, considering that these countries likely exhibit significant variations in structural variables. It is of course beyond the scope of this study, but it would be interesting to see comparative studies of SDT.

Methods:

I cannot stress enough the recommendation of putting this section before results. Besides that, I suggest declaring more clearly that not only the model of SDT but also the measures are based on Turchin’s previous work. That would limit questions to the construction of structural variables and indexes.

This section should also explain the sources used for every measure and include more information that clarifies and justifies the chosen methods. Most of this is already written in supporting information, but it should be explained to some extent in the body of the paper.

Two last important points. The data provided do not meet PLOS politics. The data points used to create the provided aggregated measures, as well as the code used to process the information, should be included. Also, the financial disclosure link is broken.

Second evaluator

The paper presents a novel methodological and theoretical proposal for the analysis of socio-political dynamics in the Chilean context. The structural demographic theory is introduced and applied to approach the process of overproduction of the Chilean government elites and its relationship with the processes of mobilization and social conjuncture that the country has experienced. This aspect is in fact an element to be highlighted since it allows us to approach an understanding of the complex phenomena of social mobilizations and contributes to the development of future research in the región.

The following are a series of formal comments to be taken into consideration by the authors:

1) While it is true that at the introductory level the authors address the Structural Demographic Theory and the way it will be applied in the study, I suggest expanding a little more in detail on the scope of this theory to the study of social dynamics. Social demographic theory provides a conceptual framework for understanding patterns of population change at different stages of a country's socioeconomic development. The authors have supported with an exhaustive level of detail the methodological guidelines to account for the dynamics of change in the Chilean context with the variables for this purpose. However, the reader may have the feeling that the complex framework that Chile has experienced in this period (1938-2019) is reduced to a brief analysis of variables, omitting the complexity of the political, social, cultural, and historical reality of the country.

2) The phenomenon of social mobilizations is in essence a complex phenomenon and obeys the particularity of a specific context and time. Several elements have been mentioned that accompany the emergence of mobilization. While it is true that the authors mention in the introductory section of the manuscript, I quote: "Therefore, SDT does not seek to encompass all the complexity associated with the phenomenon of political conflict in social systems, nor does it ignore the importance of the agency of political actors. It only allows us to predict, through the interaction of structural variables, periods when the collective mood is more likely to trigger major scenarios of institutional instability" and in fact, the study does not seek to reduce the complexity of the social reality of Chile and the social mobilizations of October 18 to an analysis of variables, I suggest that the authors expand in more detail in the introductory section these elements of discussion.

3) The manuscript is structured from an introductory section followed by a middle section that introduces the reader to the findings of the study and then the reader will find a methodological section. I suggest that the methodological section comes first before the results. The manuscript is accompanied by supplementary material that the reader can consult to go into more detail on how the analysis proceeded. Nevertheless, the material contains important information that would be worth including in the methodological section of the article. To expand on the details of the procedure, the collection of the empirical corpus, the databases consulted, the previous works consulted. It would be more convenient for the reader to find this information in the methodological section and leave the more technical information guiding the methodological elaboration of the study in the supplementary file. Equations etc.

4) Finally, I suggest to the authors to expand the discussions of the study in terms of the scope of this research for the development of future studies in the region. How the use of Demographic Structural Theory can be a conceptual framework to address regional inequalities and disparities orchestrated by the elites. In terms of the challenges for the case of Chile to agitate planning processes and public policies for the future. By understanding the stages of transition that Chile has experienced in its history, it would be possible to infer changes in the population structure and think about the possibility of adapting policies in areas such as health, education, employment, social security, etc.

General considerations of the Academic Editor:

I suggest that the following recommendations of the evaluators be prioritized

a. Is the manuscript technically sound, and do the data support the conclusions? PARTLY

b. Has the statistical analysis been performed appropriately and rigorously? NO

c. Have the authors made all data underlying the findings in their manuscript fully available? NO

d. Change the order of the sections of the article (method before results)

Reviewers' comments:

Reviewer's Responses to Questions

**Comments to the Author**

1. Is the manuscript technically sound, and do the data support the conclusions?

Reviewer #1: Yes

Reviewer #2: Partly

2. Has the statistical analysis been performed appropriately and rigorously? 

Reviewer #1: Yes

Reviewer #2: No

3. Have the authors made all data underlying the findings in their manuscript fully available?

Reviewer #1: Yes

Reviewer #2: No

4. Is the manuscript presented in an intelligible fashion and written in standard English?

Reviewer #1: Yes

Reviewer #2: Yes

5. Review Comments to the Author

Reviewer #1: The paper presents a novel methodological and theoretical proposal for the analysis of socio-political dynamics in the Chilean context. The structural demographic theory is introduced and applied to approach the process of overproduction of the Chilean government elites and its relationship with the processes of mobilization and social conjuncture that the country has experienced. This aspect is in fact an element to be highlighted since it allows us to approach an understanding of the complex phenomena of social mobilizations and contributes to the development of future research in the region, applied to other Latin American countries due to their proximity, to the extent that the so-called social outbreak of October 18, 2019 constituted an important precedent in the history of the country and the feeling of social struggle managed to energize and be reflected in contexts such as the Colombian one, which in the middle of the third peak of the pandemic, the country took to the streets in massive marches.

The following are a series of formal comments to be taken into consideration by the authors:

1) While it is true that at the introductory level the authors address the Structural Demographic Theory and the way it will be applied in the study, I suggest expanding a little more in detail on the scope of this theory to the study of social dynamics. Social demographic theory provides a conceptual framework for understanding patterns of population change at different stages of a country's socioeconomic development. The authors have supported with an exhaustive level of detail the methodological guidelines to account for the dynamics of change in the Chilean context with the variables for this purpose. However, the reader may have the feeling that the complex framework that Chile has experienced in this period (1938-2019) is reduced to a brief analysis of variables, omitting the complexity of the political, social, cultural, and historical reality of the country.

2) The use of the Demographic Structural Theory may be subject to criticism from approaches related to the theory of social mobilizations, in addition to the fact that the latter has been mostly used in the Latin American context, which have focused on human agency as a characteristic dimension of the subjects to make use of the environments and forge transformation processes. It is also necessary to highlight the place of social conflicts as an enabling dimension of the emergence of change, of the struggle against social injustices, inequality and in many cases oppression that are generated in certain classes, groups, and social movements. The political dimension is a relevant factor, insofar as the relationship between social movements and struggles develops in parallel with political processes, forging mechanisms that enable changes in the political structure. Similarly, the identity dimensions involved in social movements are fundamental factors, given that historically it has been known that a large part of social mobilizations emerge around the defense of identities such as race and ethnicity, and involve cultural and symbolic values in the processes of social struggle.

3) The phenomenon of social mobilizations is in essence a complex phenomenon and obeys the particularity of a specific context and time. Several elements have been mentioned that accompany the emergence of mobilization. While it is true that the authors mention in the introductory section of the manuscript, I quote: "Therefore, SDT does not seek to encompass all the complexity associated with the phenomenon of political conflict in social systems, nor does it ignore the importance of the agency of political actors. It only allows us to predict, through the interaction of structural variables, periods when the collective mood is more likely to trigger major scenarios of institutional instability" and in fact, the study does not seek to reduce the complexity of the social reality of Chile and the social mobilizations of October 18 to an analysis of variables, I suggest that the authors expand in more detail in the introductory section these elements of discussion.

4) The manuscript is structured from an introductory section followed by a middle section that introduces the reader to the findings of the study and then the reader will find a methodological section. I suggest that the methodological section comes first before the results. The manuscript is accompanied by supplementary material that the reader can consult to go into more detail on how the analysis proceeded. Nevertheless, the material contains important information that would be worth including in the methodological section of the article. To expand on the details of the procedure, the collection of the empirical corpus, the databases consulted, the previous works consulted. It would be more convenient for the reader to find this information in the methodological section and leave the more technical information guiding the methodological elaboration of the study in the supplementary file. Equations etc.

5) Finally, I suggest to the authors to expand the discussions of the study in terms of the scope of this research for the development of future studies in the region. How the use of Demographic Structural Theory can be a conceptual framework to address regional inequalities and disparities orchestrated by the elites. In terms of the challenges for the case of Chile to agitate planning processes and public policies for the future. By understanding the stages of transition that Chile has experienced in its history, it would be possible to infer changes in the population structure and think about the possibility of adapting policies in areas such as health, education, employment, social security, etc.

Reviewer #2: This paper presents an interesting application of Structural Demographic Theory (SDT) to the case of Chile in the last century. It mostly draws on Goldstone and Turchin’s work, but it seems to go a bit further in terms of methodology by incorporating predictive analytics. The model used achieves to predict social unrest periods, leading the authors to argue that it shows the “underlying cause of Chilean political instability”. Although I think this is a very interesting piece of work, I believe it would benefit from some changes that I would suggest hereafter:

First, I will address some form issues. I recommend modifying the title of the article to explicitly indicate that it is an application of SDT. That way it would be easier to identify the paper as a part of a broader literature, and would make it more accessible to a wider audience beyond those specifically interested in the Chilean case. Second, I suggest moving the materials and methods section before Results. Such order is common among PLOS articles and it would make a lot easier to understand the results section. Lastly, it appears that there are some issues with the equations when exporting from the text editor (see lines 73 and 348) and some equations seem to have errors (I’m guessing in eq 1 frequency and total occurrence are indexed in i).

Now I will address the contents of the paper going section by section.

Introduction:

This section seems very clear to me. It frames adequately SDT as an “interpretative framework” that “does not seek to encompass all the complexity” behind political instability but is sufficient to model (and predict) social unrest periods. In terms of the explanation of the Chilean case, it is weird that it goes into detail about social unrest during 2019, but does not address other important political instability events during the time window studied. It seems important to provide contextual information on political instability throughout the 20th century in Chile, as the study later treats at least two other events with the same level of importance as the "social outbreak" of 2019.

Results:

Once the goal of the study is introduced, the paper jumps to the results section with a visual inspection of four time series, one for each independent variable and one for an index that aggregates them with a multiplicative model. The visual inspection closes with a big conclusion: “it can be seen how the structural demographic theory predicts an increase in political tension in the Chilean social system prior to the outbreak of the main events of social instability of the last 80 years”. This is one of the main reasons why I recommend putting the methods section before. As I read this without knowing the methods that were going to be used to test predictability, it was easy to conclude the study was flawed. Visual inspection does not support predictability. I suggest either removing this initial part of the results or moderating the conclusion.

Then, the outcome of the model is presented. The authors introduce the Observed Political Instability Index (OPII) and while it can be inferred that this measure represents the dependent variable created by the authors, it would be a lot clearer if the methods section were before the results.

Finally, prediction models are presented: a non-linear regression proves that SDT can fit OPII, Arima model shows a temporal correlation between changes in observed and estimated outcomes, and forecasting provides evidence that SDT can predict the OPII using information from a year before. Unfortunately, this core aspect of the study, which addresses the research question of whether SDT can predict social instability, is described in a brief paragraph. My suggestion is to extend the explanation of these results and go deeper into them. You could go further in the explanation of the magnitude of the parameters, test if the prediction would stand for a bigger time horizon, or identify which structural variable is more important to the prediction (which appears to be an important aspect in the Discussion section)

The last section of the results shows the interaction of two of the dimensions of SDT. This is coherent with the theory that stays that there are feedback loops between the variables. It is not clear why this was the only interaction tested. I suggest stating in the introduction a clear research question that is answered by this analysis, explaining in the results section why it is the only interaction addressed, or extending this section to include tests for other interactions.

As a final comment for this section, I would like to emphasize that these results discuss predictability but do not establish causality. I strongly recommend being cautious in making any claims that imply causation has been proven.

Discussion:

The discussion gives a rich explanation of the Chilean case and why it is in line with the results of the study. Although I think this is very valuable, it would be important to mention something about the policy implications of your results, taking into account that in the introduction you stated “Understanding the structural forces that underlie the dynamics of political instability in modern societies is essential to adopt measures that allow adapting social systems prior to the irruption of events” What can a policy maker learn from your study? I would also like to see some discussion about the implication of the results to SDT.

I find the mention of social distress in other countries highly relevant, considering that these countries likely exhibit significant variations in structural variables. It is of course beyond the scope of this study, but it would be interesting to see comparative studies of SDT.

Methods:

I cannot stress enough the recommendation of putting this section before results. Besides that, I suggest declaring more clearly that not only the model of SDT but also the measures are based on Turchin’s previous work. That would limit questions to the construction of structural variables and indexes.

This section should also explain the sources used for every measure and include more information that clarifies and justifies the chosen methods. Most of this is already written in supporting information, but it should be explained to some extent in the body of the paper.

Two last important points. The data provided do not meet PLOS politics. The data points used to create the provided aggregated measures, as well as the code used to process the information, should be included. Also, the financial disclosure link is broken.

6. PLOS authors have the option to publish the peer review history of their article (what does this mean?). If published, this will include your full peer review and any attached files.

Reviewer #1: No

Reviewer #2: No

---

## [Author Response · Author response to Decision Letter 0]

29 Jan 2024

First reviewer: 

Comment 1.1: 

This paper presents an interesting application of Structural Demographic Theory (SDT) to the case of Chile in the last century. It mostly draws on Goldstone and Turchin’s work, but it seems to go a bit further in terms of methodology by incorporating predictive analytics. The model used achieves to predict social unrest periods, leading the authors to argue that it shows the “underlying cause of Chilean political instability”. Although I think this is a very interesting piece of work, I believe it would benefit from some changes that I would suggest hereafter: 

Answer 1.1 

We greatly appreciate the interest shown by the reviewer regarding the theoretical and methodological approach employed in our research to address the study of political instability dynamics in Chile. 

At the same time, in response to the suggestions received, in this current version, we have refined our conclusions, aiming to make it clear that the model used enables the prediction and reconstruction of the observed dynamics of Chilean political instability (1938-2018), explaining an acceptable percentage of the variability. However, our empirical analyses do not imply the existence of causality. In this regard, in the Discussion section, we have verbatim incorporated the following phrase: 

“Although the results presented in this section are consistent with the predictions derived from adapting the SDT to the Chilean case, they do not empirically establish causality” 

Comment 1.2: 

First, I will address some form issues. 

I recommend modifying the title of the article to explicitly indicate that it is an application of SDT. That way it would be easier to identify the paper as a part of a broader literature, and would make it more accessible to a wider audience beyond those specifically interested in the Chilean case. 

Answer 1.2: 

Thank you for your suggestion. Based on this feedback, we have decided to modify the title to: 

“Squeezed from the top: “Social Outburst” (2019) and elite overproduction. A study of the dynamics of Chilean political instability from the approach of Structural Demographic Theory”. 

In this manner, the prospective reader can identify the article with a broader theoretical framework that has been utilized in the past to address instances of political instability in other societies. 

Comment 1.3: 

Second, I suggest moving the materials and methods section before Results. Such order is common among PLOS articles and it would make a lot easier to understand the results section. 

Answer 1.3: 

We accept this suggestion. In the new version, we have incorporated the materials and methods section before presenting the results. Indeed, this modification enhances the readability and comprehensibility of the work. 

Comment 1.4: 

Lastly, it appears that there are some issues with the equations when exporting from the text editor (see lines 73 and 348) and some equations seem to have errors (I’m guessing in eq 1 frequency and total occurrence are indexed in i). 

Answer1.4: 

Thank you very much for the observation. We apologize for any formality issues in the text. These errors may have arisen from the use of multiple Office versions by the authors who edited the text. In this current version, we have rectified the errors in the equations. 

Comment 1.5: 

Introduction: 

This section seems very clear to me. It frames adequately SDT as an “interpretative framework” that “does not seek to encompass all the complexity” behind political instability but is sufficient to model (and predict) social unrest periods. In terms of the explanation of the Chilean case, it is weird that it goes into detail about social unrest during 2019, but does not address other important political instability events during the time window studied. It seems important to provide contextual information on political instability throughout the 20th century in Chile, as the study later treats at least two other events with the same level of importance as the "social outbreak" of 2019. 

Answer 1.5: 

Our research aims to emphasize the Social Outburst (2019), which represents the most significant disruption that Chilean institutions have faced since the return to democracy in the 1990s. This event of social instability, with significant repercussions on the country's current political landscape, serves as a paradigmatic case. We seek to understand how, in a nation that was considered an example of economic stability and institutional strength, a seemingly minor event (a 30-peso increase in public transportation fares) could trigger a crisis of such magnitude in governance. This was our primary motivation. 

Numerous studies have addressed this question, conducting comprehensive analyses of the social, economic, political, and cultural factors that may have influenced the emergence of this social phenomenon. However, to the best of our knowledge, few empirical studies have approached this issue by reconstructing the dynamics of long-term internal tensions within Chilean society. These tensions can contribute to an understanding of the current political scenario in the country. Hence, the analysis of the "Social Outburst", through the study of medium and long-term pressures on the three structural variables, holds a central role in our perspective and in this present research. 

Notwithstanding the above, we have decided to heed the reviewer's suggestion. In this regard, we have included the following paragraph in the introduction: 

“In the last 80 years, Chile has experienced other events of political instability of great relevance. The most important of them, the crisis of the late 60s and early 70s, which culminated with a Coup d'état (1973) that represented a profound break with the political and economic model that prevailed until then. This event meant the installation of a dictatorial regime (1973-1989) that was responsible for the death and disappearance of more than 3,000 victims. On the other hand, during the Latin American debt crisis (beginning of the 80s), an important cycle of civil mobilizations erupted in Chile, which constituted the main challenge faced by the Pinochet dictatorship.” 

(In the version with tracked changes, these are highlighted with "Comment 1.5”) 

Comment 1.6: 

Results: 

Once the goal of the study is introduced, the paper jumps to the results section with a visual inspection of four time series, one for each independent variable and one for an index that aggregates them with a multiplicative model. The visual inspection closes with a big conclusion: “it can be seen how the structural demographic theory predicts an increase in political tension in the Chilean social system prior to the outbreak of the main events of social instability of the last 80 years”. This is one of the main reasons why I recommend putting the methods section before. As I read this without knowing the methods that were going to be used to test predictability, it was easy to conclude the study was flawed. Visual inspection does not support predictability. I suggest either removing this initial part of the results or moderating the conclusion. 

Answer 1.6: 

Thank you very much for the suggestion. In this new version of the manuscript, we have placed the materials and methods section before the results section. This approach allows for a clearer presentation of the primary research findings and enhances the comprehension of the analyses conducted. 

Indeed, as suggested by the reviewer, in Figure 1, we only depict the dynamics of the three structural variables (SFD, MMP, and EMP) and the PSI indicator. Additionally, we have identified the three primary sociopolitical instability events in Chile over the past 80 years: the Coup d'état (1973), popular mobilizations (1980s), and the "Social Outburst " (2019). This figure is intended solely for visual inspection. 

Therefore, in this new version of the manuscript, we limit ourselves to qualitatively describing the temporal changes in each of the presented variables. Simultaneously, we describe that prior to the emergence of each of the major sociopolitical instability events, an upward trajectory in the PSI indicator can be observed. 

In this way, the phrase: 

“Figure 1d shows the dynamics of the PSI index, in which it can be seen how the structural demographic theory predicts an increase in political tension in the Chilean social system prior to the outbreak of the main events of social instability of the last 80 years” 

It is replaced by: 

“Figure 1d shows that the periods in which an upward trajectory of the PSI indicator is obtained, which according to SDT is indicative of an increase in political stress, coincide temporally with the occurrence of major events of social instability in the last 80 years”. 

(In the version with tracked changes, these are highlighted with "Comment 1.6”) 

Comment 1.7: 

Then, the outcome of the model is presented. The authors introduce the Observed Political Instability Index (OPII) and while it can be inferred that this measure represents the dependent variable created by the authors, it would be a lot clearer if the methods section were before the results. 

Answer 1.7: 

As recommended by the reviewer, the observed political instability index was constructed by us, as described in detail in the methodology section. To prevent any confusion, in this new version of the manuscript, the materials and methods section precedes the presentation of the results. 

Comment 1.8: 

Finally, prediction models are presented: a non-linear regression proves that SDT can fit OPII, Arima model shows a temporal correlation between changes in observed and estimated outcomes, and forecasting provides evidence that SDT can predict the OPII using information from a year before. Unfortunately, this core aspect of the study, which addresses the research question of whether SDT can predict social instability, is described in a brief paragraph. My suggestion is to extend the explanation of these results and go deeper into them. You could go further in the explanation of the magnitude of the parameters, test if the prediction would stand for a bigger time horizon, or identify which structural variable is more important to the prediction (which appears to be an important aspect in the Discussion section). 

Answer 1.8: 

Thank you very much for the recommendations. Based on these suggestions, in the current version, two additional analyses were conducted. 

Comparison of alternative models (using different possible combinations of the product of the three structural variables). Detailed results, including the model, parameter values, goodness of fit, and AIC values, have been incorporated into a new table in the supplementary materials (S1.File: S1.Table). 

Cross-validation analysis using projections of 2, 4, 6, and 8 steps ahead. The results obtained are shown in a new figure included in the supplementary material (S1.File: S3Figure). 

Considering these analyses, explanations in the results and discussion sections have been expanded (in the version with tracked changes, these are highlighted with "Comment 1.8"). 

Regarding the analyses mentioned above, the following text has been added to: 

 Results section: 

Comparison of alternative models: 

“The non-linear regression analysis (NLS) indicates the existence of a statistically significant relationship between the PSI* indicator and the observed political instability dynamics (pseudo-R2 = 0.55; parameter values: a=0.54, p-value=1.86 e-11*; b=0.36 , p-value =2.60e-08*; c=0.07 , p-value=0.421). Alternatively, all possible combinations derived from the main model (PSI* = SFD a x MMP b X EMP c) were evaluated, considering the product of 2 components and each of the individual components separately (S1.File: S1.Table). The results obtained show that the best model in terms of fit and complexity is PSI* ( pseudo-R2= 0.55, AIC= -67.9). However, it is important to highlight that the alternative model SFDa x MMPb also presents an important relationship with the observed political instability dynamics (pseudo-R2=0.53, AIC=-69.2)" 

Cross-validation analysis: 

“The regression with ARIMA errors model between the rate of change of PSI* (see methodology) and the rate of change of the observed political violence index shows a positive linear relationship between the two indicators (β=1.25. CI-95%: 0.57-1.93, p-value=0.0005) (Fig. 2b). On the other hand, the predictions obtained when using the PSI* model, by the cross-validation technique and the projection one step forward (k-1), are shown in Figure 4c. It shows an important correspondence between the dynamics of political instability observed and that predicted through the SDT model (RMSE= 0.15) (Fig. 2c). Even, this strong correspondence is maintained by increasing the prediction window of the model several steps forward (2,4,6 and 8 steps). In this case, although the prediction noise increases, the dispersion levels of the residual values (RMSE) still show a good fit between the political instability predicted by the model and the observed political instability index (S1.File:S3Figure). These results seem to indicate that structural demographic theory (SFT) effectively predicts the main periods of increased social tension and political instability in Chilean society over the last 80 years.” 

Discussion section: 

Comparison of alternative models: 

“Our results show that the SDT predicts an increase in political tension in Chilean society prior to the emergence of the main social instability events of the last 80 years. By analyzing the qualitative dynamics of each of the 3 components of the PSI indicator (SFD, MMP, and EMP), it is possible to recognize some general trends. All events of political instability are preceded by an increase in tension in the popular sectors. This observation is corroborated in part by the high fitted values of the alternative model (SFDa x MMPb) when compared to the trajectory of the observed dynamics of the political instability index. At the same time, prior to the coup d'état (1973) and the cycle of mobilizations in the 1980s, significant increases were observed in one of the two indicators used as a proxy for SFD, as well as a more moderate increase from the beginning of the 2010s. Regarding the elites, tension in this group increased prior to the coup of 1973 and the social outburst of 2019, while in the 1980s it showed a downward trajectory. 

In this sense, according to our estimates, the increase in tension among the elites precedes the events of political conflict that led to profound changes in the institutional regime (the coup d'état of 1973) or in processes of opening to potential reforms (the Social Outburst of 2019). On the other hand, in periods of accumulation of fiscal stress and increased pressure on the general population without accompanying of elite mobilization potential, although scenarios of social conflict emerged (mobilizations of the 1980s), they did not lead to institutional changes. Therefore, the above observations could suggest the potential existence of a correspondence between the dynamics of structural variables and certain characteristics of the degree of magnitude and probability of success of emerging conflict events. With the information available to date, it is too early to draw a conclusion in this regard. Future studies focused on the reconstruction of the 3 structural variables in different contemporary societies will allow us to go deeper into these aspects.” 

Cross-validation analysis: 

“Despite its relative simplicity, the political stress index (PSI) predicted by the structural-demographic model captured the occurrence of social conflict and political vulnerability by predicting levels of discontent in different social groups. The model is well-adjusted in spite of the profound changes experienced in political and economic institutions in the last 80 years (1,2). This adjustment capacity is sustained even when using multi-step forward predictive methodologies (up to 8 years). In this sense, our results indicate that, at least within the period of interest, monitoring the dynamics of the 3 structural variables allows us to anticipate the state of stability of the institutional system. Future studies could advance conceptual frameworks that allow the integration of SDT with other approaches, which would contribute to a better understanding of the complexity of these social phenomena and how the vulnerability predicted by structural demographic theory combines with other factors to open the opportunity for the mobilization of different social groups.” 

(In the version with tracked changes, these are highlighted with "Comment 1.8”) 

Comment 1.9: 

The last section of the results shows the interaction of two of the dimensions of SDT. This is coherent with the theory that stays that there are feedback loops between the variables. It is not clear why this was the only interaction tested. I suggest stating in the introduction a clear research question that is answered by this analysis, explaining in the results section why it is the only interaction addressed, or extending this section to include tests for other interactions. 

Answer 1.9: 

In response to this suggestion, we have decided to add an explanation in the introduction to justify the emphasis on the interaction between the components of the MMP and EMP variables. 

“According to the theoretical framework of SDT, one of the main drivers of sociopolitical instability is the anti-phase dynamic between the growth in the number of people who aspire to occupy elite positions and the proxies for the well-being of the population. Different empirical studies have found that there is a greater probability of the emergence of events of political violence when the well-being of the population is low and there is an overproduction of elites (a number greater than what a given society can sustain) (2-4). In this sense, exploring the relationship between these two variables (in a long wave) can contribute to better understanding the processes involved in increasing social tension.” 

(In the version with tracked changes, these are highlighted with "Comment 1.9”) 

Comment 1.10: 

As a final comment for this section, I would like to emphasize that these results discuss predictability but do not establish causality. I strongly recommend being cautious in making any claims that imply causation has been proven. 

Answer 1.10: 

We accept the suggestion made by the reviewer. From our perspective, Structural Demographic Theory presents a robust empirical model, grounded in a solid conceptual framework, which allows for predicting the dynamics of tension accumulation within a given social system, recognizing periods where there is a higher probability of the emergence of political conflict events. While it aids in understanding certain structural pressures in the medium and long term that are relevant to understanding specific sociopolitical conflict events, it does not provide sufficient elements for evaluating causality. 

In line with the above, we have incorporated some modifications in the discussion text to avoid confusion regarding our conclusions and to be more emphatic on this point. 

“Our results show that in the case of Chilean society over the last 80 years, major periods of political conflict were preceded by scenarios in which the growth of certain social groups exceeded their accessibility to resources. These processes tend to increase internal competition, affecting the relationship between elites, popular groups, and the State, impacting the stability of the institutional system (24). In this sense, the rising dynamics of the PSI political stress indicator seem to represent the periods in which the Chilean institutional system is more vulnerable to experiencing the emergence of major political conflict events. 

Although the results presented are consistent with the predictions derived from the adaptation of SDT to the Chilean case, they do not empirically establish causality. The outbreak and outcome of a political instability event depend on multiple factors that are beyond the theoretical framework of SDT and require other types of approaches that emphasize organizational, ideological, and strategic aspects (i.e., the existence of charismatic leaders, ideological identification processes, international pressures, organizational capacity, and linkage of elites with popular groups). Nevertheless, our results suggest that for the Chilean case, the SDT model provides good probabilistic estimates of the stability of the institutional system.” 

(In the version with tracked changes, these are highlighted with "Comment 1.10”) 

Comment 1.11: 

Discussion: 

The discussion gives a rich explanation of the Chilean case and why it is in line with the results of the study. Although I think this is very valuable, it would be important to mention something about the policy implications of your results, taking into account that in the introduction you stated “Understanding the structural forces that underlie the dynamics of political instability in modern societies is essential to adopt measures that allow adapting social systems prior to the irruption of events” What can a policy maker learn from your study? I would also like to see some discussion about the implication of the results to SDT. 

I find the mention of social distress in other countries highly relevant, considering that these countries likely exhibit significant variations in structural variables. It is of course beyond the scope of this study, but it would be interesting to see comparative studies of SDT. 

Answer 1.11: 

Thanks to the reviewer for pointing us in this direction. One of the main implications of the SDT is that political instability, as configured in contemporary societies, is driven by the combined effects of a series of structural factors, which are not quickly addressed through short-term policy innovations/change. As a result, we proceeded in two ways. First, we amended the sentence in the introduction in ways better aligned with our argument's main trust. That sentence now reads as follows: 

“Understanding the factors that underlie the dynamics of political instability in modern societies is essential to identifying the structural underpinnings (and thus, long-term drivers) of contemporary political events associated with the crisis of contemporary liberal democracies.” 

Second, we included the following sentence in closing the discussion of the Chilean case: 

 "Unfortunately, this is not a trend for which we can foresee any significant changes in the near term. A crucial implication of our study is that policymakers and pro-democratic forces interested in the long-term prospects of democracy should thus devise a new "social pact" that can better counteract the structural forces (growing inequality; an overproduction of aspiring elites) that induce democratic crises in contemporary liberal regimes. Such “social pact" should include traditional social policy targets such as health, education, employment, and social security and innovative ways of tackling inequality and the social mobility expectations of new generations. Moreover, the “social pact” should be compatible with a new era in which pursuing ecologically sustainable economic growth is inescapable.” 

(In the version with tracked changes, these are highlighted with "Comment 1.11”) 

Comment 1.12: 

Methods: 

I cannot stress enough the recommendation of putting this section before results. Besides that, I suggest declaring more clearly that not only the model of SDT but also the measures are based on Turchin’s previous work. That would limit questions to the construction of structural variables and indexes. 

This section should also explain the sources used for every measure and include more information that clarifies and justifies the chosen methods. Most of this is already written in supporting information, but it should be explained to some extent in the body of the paper. 

Answer 1.12: 

Thank you for all the feedback. In the current version of the manuscript, we have presented the Materials and Methods section before the results. Furthermore, we have incorporated more detail regarding the sources used, the analyses conducted, and the data management. 

Comment 1.13: 

Two last important points. The data provided do not meet PLOS politics. The data points used to create the provided aggregated measures, as well as the code used to process the information, should be included. Also, the financial disclosure link is broken. 

Answer 1.13: 

In the current version of the manuscript, we have made all the data used for the estimation of the three structural variables available. We have also included the data related to the construction of the observed political instability index. Additionally, in the supplementary database file, we have included information about the sources and a link for accessing the digital databases. Furthermore, we have made the R analysis codes used in the study available for access. 

Second reviewer: 

Comment 2.1: 

The paper presents a novel methodological and theoretical proposal for the analysis of socio-political dynamics in the Chilean context. The structural demographic theory is introduced and applied to approach the process of overproduction of the Chilean government elites and its relationship with the processes of mobilization and social conjuncture that the country has experienced. This aspect is in fact an element to be highlighted since it allows us to approach an understanding of the complex phenomena of social mobilizations and contributes to the development of future research in the region. 

Answer 2.1: 

We are glad to hear that the reviewer showed interest in our research and its potential future implications. 

Comment 2.2: 

The following are a series of formal comments to be taken into consideration by the authors: 

1)While it is true that at the introductory level the authors address the Structural Demographic Theory and the way it will be applied in the study, I suggest expanding a little more in detail on the scope of this theory to the study of social dynamics. Social demographic theory provides a conceptual framework for understanding patterns of population change at different stages of a country's socioeconomic development. The authors have supported with an exhaustive level of detail the methodological guidelines to account for the dynamics of change in the Chilean context with the variables for this purpose. However, the reader may have the feeling that the complex framework that Chile has experienced in this period (1938-2019) is reduced to a brief analysis of variables, omitting the complexity of the political, social, cultural, and historical reality of the country. 

Answer 2.2: 

We appreciate the reviewer for their suggestion. Based on this comment, we have made some additions to the original text with the purpose of emphasizing that our objective is not to encompass all the complexity, or all the factors associated with the emergence of social conflict processes and events. 

In the discussion, we have included the following lines: 

“Although the results presented are consistent with the predictions derived from the adaptation of SDT to the Chilean case, they do not empirically establish causality. The outbreak and outcome of a political instability event depend on multiple factors that are beyond the theoretical framework of SDT and require other types of approaches that emphasize organizational, ideological, and strategic aspects (i.e., the existence of charismatic leaders, ideological identification processes, international pressures, organizational capacity, and linkage of elites with popular groups). Nevertheless, our results suggest that for the Chilean case, the SDT model provides good probabilistic estimates of the stability of the institutional system.” 

“Despite its relative simplicity, the political stress index (PSI) predicted by the structural-demographic model captured the occurrence of social conflict and political vulnerability by predicting levels of discontent in different social groups. The model is well-adjusted in spite of the profound changes experienced in political and economic institutions in the last 80 years (1,2). This adjustment capacity is sustained even when using multi-step forward predictive methodologies (up to 8 years). In this sense, our results indicate that, at least within the period of interest, monitoring the dynamics of the 3 structural variables allows us to anticipate the state of stability of the institutional system. Future studies could advance conceptual frameworks that allow the integration of SDT with other approaches, which would contribute to a better understanding of the complexity of these social phenomena and how the vulnerability predicted by structural demographic theory combines with other factors to open the opportunity for the mobilization of different social groups.” 

(In the version with tracked changes, these are highlighted with "Comment 2.2”) 

The social, political, and cultural complexities associated with the 1973 coup, the cycle of mobilizations in the 1980s, and the 2019 Social Outburst have been extensively and comprehensively addressed by many previous works (Góngora, 1981; Stabili, 2003; Salazar, 2015; Garretón et al., 2016; Fischer, 2017; Rodríguez, 2018; Correa, 2021; Somma et al., 2021; Castro-Abril et al., 2021). In this sense, our work does not seek to disregard these complexities or engage in confrontation with them. Our objective is to contribute to the understanding of these social phenomena using an approach that has not been used before to study sociopolitical conflict in Chile or other Latin American countries. Structural Demographic Theory aims to describe the medium- and long-term trends and dynamics related to structural tensions that lead to a certain social system experiencing periods of greater and lesser vulnerability to the emergence of significant sociopolitical instability events. 

It is important to note that from the perspective of Structural Demographic Theory (SDT), human societies are complex systems where sociopolitical instability events constitute emergent processes that recur over time and follow a relatively periodic trend (when studied over very long-time scales, which is not the case in this study). This empirical observation (Goldstone, 1991; Turchin & Hall, 2003; Turchin, 2012, 2013, 2016; Turchin and Korotayev, 2020) seems to suggest the existence of underlying "forces" that transcend institutional (political and economic) and cultural differences between different societies. These "underlying forces" should not be interpreted as the "ultimate cause" of social conflict events. They merely represent impulses related to the accumulation of tensions in the medium and long term within a particular social system, allowing for the probabilistic definition of temporal spaces where there is a higher propensity for social conflict events to emerge. These events result from a multitude of factors specific to each historical context. 

In order to contribute to the understanding of the dynamics of tension accumulation in a society over the medium and long term, authors like Goldstone and Turchin propose the Structural Demographic Model, which is based on some processes that occur at the population (and social group) and macroeconomic levels (fiscal pressure). This approach, resulting in the construction of a relatively simple model, aims to capture the essence of these historical impulses, attempting to explain as much of the variance as possible and contributing to the interpretation and partial prediction of the phenomenon of interest. 

The simplicity of the model should not be understood as an underestimation of the complexity of the phenomenon being studied. On the contrary, the aim of such approaches is to test certain variable interactions, grounded in a solid theoretical framework, that allows for generating an approximate understanding of these complex phenomena. This enables the drawing of some conclusions that, while limited, are manageable and have practical utility. 

Comment 2.3: 

 2) The phenomenon of social mobilizations is in essence a complex phenomenon and obeys the particularity of a specific context and time. Several elements have been mentioned that accompany the emergence of mobilization. While it is true that the authors mention in the introductory section of the manuscript, I quote: "Therefore, SDT does not seek to encompass all the complexity associated with the phenomenon of political conflict in social systems, nor does it ignore the importance of the agency of political actors. It only allows us to predict, through the interaction of structural variables, periods when the collective mood is more likely to trigger major scenarios of institutional instability" and in fact, the study does not seek to reduce the complexity of the social reality of Chile and the social mobilizations of October 18 to an analysis of variables, I suggest that the authors expand in more detail in the introductory section these elements of discussion. 

Answer 2.3: 

Following this suggestion, we have decided to incorporate the following lines toward the end of the discussion: 

“Politicians and political pundits often interpret events like Chile's "Social Outburst" as the result of contingent agency by specific groups and their leadership. For instance, in 2019, the Chilean government attributed the riots to a plan for political destabilization perpetrated by extreme leftist groups and their international allies. Our results suggest, instead, that the events of 2019 correspond to a "collective action cascade" triggered by contingent events but enabled by an underlying set of structural factors. Given that Chile is often considered a development model for other Latin American countries, our results further point to possible structural weaknesses of such a predicament. On this note, future works might seek to extend the application of the SDT to other instances of political instability in the region to test the external validity of our findings.” 

(In the version with tracked changes, these are highlighted with "Comment 2.3”) 

Comment 2.4: 

3) The manuscript is structured from an introductory section followed by a middle section that introduces the reader to the findings of the study and then the reader will find a methodological section. I suggest that the methodological section comes first before the results. The manuscript is accompanied by supplementary material that the reader can consult to go into more detail on how the analysis proceeded. Nevertheless, the material contains important information that would be worth including in the methodological section of the article. To expand on the details of the procedure, the collection of the empirical corpus, the databases consulted, the previous works consulted. It would be more convenient for the reader to find this information in the methodological section and leave the more technical information guiding the methodological elaboration of the study in the supplementary file. Equations etc. 

Answer 2.4: 

In the current version of the manuscript, we have presented the Materials and Methods section before the results. Furthermore, we have incorporated more detail regarding the sources used, the analyses conducted, and the data management. 

Comments 2.5: 

4) Finally, I suggest to the authors to expand the discussions of the study in terms of the scope of this research for the development of future studies in the region. How the use of Demographic Structural Theory can be a conceptual framework to address regional inequalities and disparities orchestrated by the elites. In terms of the challenges for the case of Chile to agitate planning processes and public policies for the future. By understanding the stages of transition that Chile has experienced in its history, it would be possible to infer changes in the population structure and think about the possibility of adapting policies in areas such as health, education, employment, social security, etc. 

Answer 2.5: 

Thank you very much for all the suggestions. In response to this comment, we incorporate the following paragraphs into the discussion: 

Same as in comment 2.3: 

“Politicians and political pundits often interpret events like Chile's "Social Outburst" as the result of contingent agency by specific groups and their leadership. For instance, in 2019, the Chilean government attributed the riots to a plan for political destabilization perpetrated by extreme leftist groups and their international allies. Our results suggest, instead, that the events of 2019 correspond to a "collective action cascade" triggered by contingent events but enabled by an underlying set of structural factors. Given that Chile is often considered a development model for other Latin American countries, our results further point to possible structural weaknesses of such a predicament. On this note, future works might seek to extend the application of the SDT to other instances of political instability in the region to test the external validity of our findings.” 

And 

“A crucial implication of our study is that policymakers and pro-democratic forces interested in the long-term prospects of democracy should thus devise a new "social pact" that can better counteract the structural forces (growing inequality; an overproduction of aspiring elites) that induce democratic crises in contemporary liberal regimes. Such “social pact" should include traditional social policy targets such as health, education, employment, and social security and innovative ways of tackling inequality and the social mobility expectations of new generations. Moreover, the “social pact” should be compatible with a new era in which pursuing ecologically sustainable economic growth is inescapable” 

(In the version with tracked changes, these are highlighted with "Comment 2.5”) 

General considerations of the Academic Editor: 

Comment 1: 

I suggest that the following recommendations of the evaluators be prioritized 

a. Is the manuscript technically sound, and do the data support the conclusions? PARTLY 

b. Has the statistical analysis been performed appropriately and rigorously? NO 

c. Have the authors made all data underlying the findings in their manuscript fully available? NO 

d. Change the order of the sections of the article (method before results) 

Answer to academic editor: 

In this version of the manuscript, we have incorporated each and every comment and suggestion made by the reviewers. As a result, we believe that the current manuscript is a significantly improved version compared to the previous one. 

We have changed the order of the manuscript sections, placing the Materials and Methods before the Results, which enhances readability and improves the presentation and comprehension of the obtained results. We have expanded both the Results and the Discussion sections by conducting additional statistical analyses as suggested by the reviewers. At the same time, we have made changes to the introduction to better justify the scope and objectives of our research. Additionally, we have modified the discussion section to highlight the limitations of the theoretical framework used and to provide some general recommendations. 

The Materials and Methods section has been expanded, providing detailed information about all the sources and methodologies used. Additionally, in the supplementary files, we have included all the data used for the analyses, as well as the R code employed in this research. 

Additional modifications: 

In the time elapsed since the first draft of the manuscript was prepared, there have been updates to the databases generated by Chilean government agencies. As a result, new databases have been used for the components of the SFD variable. This has led to some changes in the statistical parameters and model fitting. The details of these modifications are provided in the version that includes the tracking of corrections. 

Additionally, with the aim of enhancing the visualization of the results, we have decided to redesign Figure 3 from the previous version and split it into two figures. Therefore, this new version includes an additional figure compared to the previous one.

---

## [Editor Report · Decision Letter 1]

5 Feb 2024

Squeezed from the top: “Social Outburst” (2019) and elite overproduction. A study of the dynamics of Chilean political instability from the approach of Structural Demographic Theory.

PONE-D-23-15713R1

Dear Dr. Muñoz-Rodríguez,

We’re pleased to inform you that your manuscript has been judged scientifically suitable for publication and will be formally accepted for publication once it meets all outstanding technical requirements.

Kind regards,

Julio Cesar Ossa, Ph.D

Academic Editor

PLOS ONE
---

## [Editor Report · Acceptance letter]

1 Apr 2024

PONE-D-23-15713R1 

PLOS ONE

Dear Dr. Muñoz-Rodríguez, 

I'm pleased to inform you that your manuscript has been deemed suitable for publication in PLOS ONE. Congratulations! Your manuscript is now being handed over to our production team.

Kind regards, 

on behalf of

Dr. Julio Cesar Ossa 

Academic Editor

PLOS ONE